# Real-Time Privacy Preservation for Robot Visual Perception

**Minkyu Choi**[1*]**, Yunhao Yang**[1*]**, Neel P. Bhatt**[1*]**, Kushagra Gupta** [1]**, Sahil Shah**[1]**, Aditya Rai**[1]**,
David Fridovich-Keil**[1]**, Ufuk Topcu**[1]**, Sandeep P. Chinchali**[1]
[1] *The University of Texas at Austin*

**Reviewed on OpenReview:** *https://openreview.net/forum?id=uMf2vn8396*

## Abstract

Many robots (e.g., iRobot's Roomba) operate based on visual observations from live video streams, and such observations may inadvertently include privacy-sensitive objects, such as personal identifiers. Existing approaches for preserving privacy rely on deep learning models, differential privacy, or cryptography. They lack guarantees for the complete concealment of all sensitive objects. Guaranteeing concealment requires post-processing techniques and thus is inadequate for real-time video streams. We develop a method for privacy-constrained video streaming, `PCVS`, that conceals sensitive objects within real-time video streams. `PCVS` takes a logical specification constraining the existence of privacy-sensitive objects, e.g., never show faces when a person exists. It uses a detection model to evaluate the existence of these objects in each incoming frame. Then, it blurs out a subset of objects such that the existence of the remaining objects satisfies the specification. We then propose a conformal prediction approach to (i) establish a theoretical lower bound on the probability of the existence of these objects in a sequence of frames satisfying the specification and (ii) update the bound with the arrival of each subsequent frame. Quantitative evaluations show that `PCVS` achieves over 95 percent specification satisfaction rate in multiple datasets, significantly outperforming other methods. The satisfaction rate is consistently above the theoretical bounds across all datasets, indicating that the established bounds hold. Additionally, we deploy `PCVS` on robots in real-time operation and show that the robots operate normally without being compromised when `PCVS` conceals objects.

## 1 Introduction

While robots utilize visual observations from video streams during operational routines for decision-making purposes, recording and disseminating such videos potentially exposes private information (Kuehne et al., 2011), raising ethical and legal concerns. These concerns include risks of the inadvertent capture of sensitive personal data, unauthorized access, and misuse of recorded footage. A recent story highlighting a Roomba taking images of a person in a toilet room attests to the legitimacy of privacy concerns during robotic operations (Guo, 2024).

Existing approaches protect privacy by concealing sensitive objects, but they either fail to guarantee complete concealment or cannot process real-time video streams. While concealing sensitive objects requires detecting and locating such objects, existing works rely on deep-learning models for object detection (Padmanabhan et al., 2023; Sugianto et al., 2024; Kagan et al., 2023). However, due to their black-box nature, deep-learning models cannot provide theoretical guarantees on the correctness of the detection results. On the other hand, formal methods techniques, such as model checking, can guarantee that a given video adheres to privacy concerns (Umili et al., 2022; Yang et al., 2023; Choi et al., 2024; Shah et al., 2025; Sharan et al., 2025; Choi et al., 2025). However, the computational complexity of formal methods techniques grows with the video length, hence they are incapable of being applied to real-time video streams.

---
*Equal Contribution

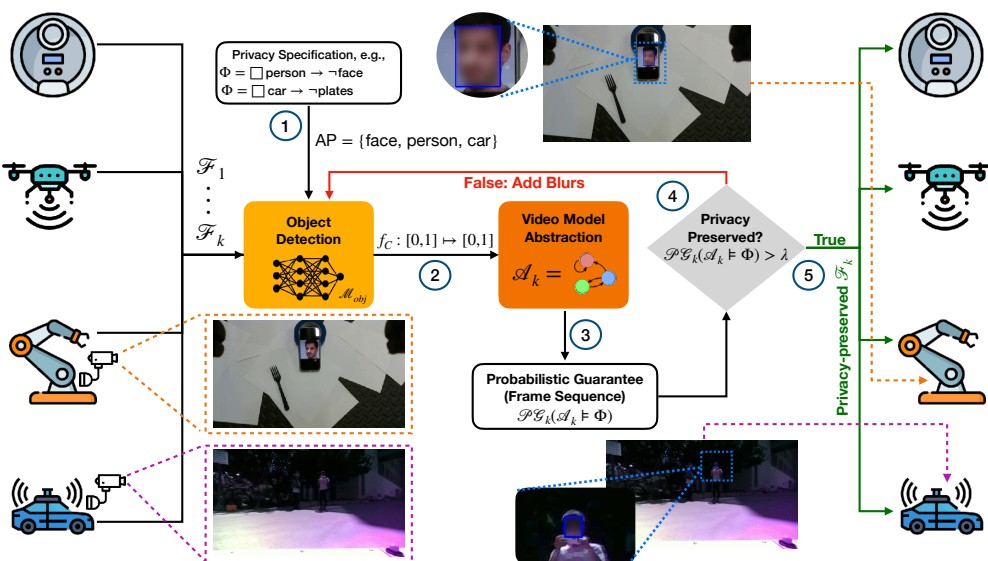

Figure 1: **Pipeline of Privacy-Constrained Video Streaming: (Step 1)** Given a privacy specification $\Phi$, we define a set $AP$ of atomic propositions describing privacy-sensitive objects. **(Step 2)** Given an incoming frame $\mathcal{F}_k$ from the video, the method uses an object detection model to detect sensitive objects in the frame. Each detection is associated with a confidence score from the object detection model. The method calibrates a confidence score to a per-frame probability bound for correct detection via a calibration function $f_C$, as in Equation 1. **(Step 3)** The method builds an abstract model $\mathcal{A}_k$ representing object detections and their probability bounds in the frame sequence $\mathcal{F}_1, ..., \mathcal{F}_k$ via Algorithm 1. Then, it computes a theoretical bound for the probability of $\mathcal{A}_k$ satisfying $\Phi$, i.e., a probabilistic guarantee $\mathcal{PG}_k(\mathcal{A}_k \models \Phi)$ using Equation 3. **(Step 4)** If $\mathcal{PG}_k(\mathcal{A}_k \models \Phi)$ is below a user-given privacy threshold $\lambda$, the method removes a subset of sensitive objects and goes back to Step 2 to recompute a guarantee. **(Step 5)** If $\mathcal{PG}_k(\mathcal{A}_k \models \Phi)$ is above $\lambda$, the method adds $\mathcal{F}_k$ back to the stream and proceeds to Step 1 with the next incoming frame. We number each step in blue.

We develop a method to conceal privacy-sensitive objects in real-time video streams from robot cameras. The method takes a logical specification constraining the existence of sensitive objects. The specifications allow users to describe complex privacy requirements with conjunctions, disjunctions, implications, etc. For each incoming frame from the video streams, the method first uses an object detection model to detect and locate all sensitive objects. Next, it removes a subset of objects (adds Gaussian blurs or blackout) so that the existence of the remaining objects satisfies the specification.

We then establish a theoretical bound on the probability of complete concealment of sensitive objects in a video stream. As deep learning models are typically over-confident in detecting objects, we use conformal prediction to calibrate the model's confidence to a probability of correct detection. Next, we express the specification as a temporal logic formula and build a finite automaton representing the object detections in a sequence of frames and the probabilities of correct detections. Then, we compute the probability that this automaton satisfies the specification (i.e., the video frames encountered so far preserve privacy).

We develop a video abstraction algorithm that allows us to optimize the computational complexity involved with the arrival of each subsequent frame from the video stream. This abstraction is key to our method achieving real-time performance, updating the probability with each frame arrival. This probability acts as a metric and helps users determine whether to use the video based on their privacy tolerance.

We evaluate the method over two large-scale datasets and present real-robot examples for real-time privacy protection. The method achieves 80 to 97 percent specification satisfaction rates in various scenarios, significantly outperforming existing automated solutions. Meanwhile, the method preserves all non-sensitive information. By seamlessly integrating concealment capabilities into the robot's visual perception system, we prevent potential privacy leakage from the robot. Simultaneously, this integration ensures the unhindered functionality of the robot's control policies, enabling it to operate normally without compromise.

## 2 Related Work

**Deep Learning for Video Privacy**   Privacy preservation in real-time video analytics has been the focus of several recent methods (Padmanabhan et al., 2023; Sugianto et al., 2024; Kagan et al., 2023; Chu et al., 2013; Wickramasuriya et al., 2005; Wang et al., 2017; Neff et al., 2019; Yuan et al., 2020; Upmanyu et al., 2009). However, they rely solely on deep learning models for object detection, i.e., detecting and blurring privacy-sensitive entities in video. Due to the black-box nature of neural networks and the lack of statistical/formal analysis, these methods lack a quantitative guarantee.

**Formal Methods for Video Privacy**   To this end, formal verification approaches have guaranteed that a given complete video adheres to privacy safety concerns formulated as temporal logical specifications. For example, recent works (Umili et al., 2022; Yang et al., 2023; Choi et al., 2024; Cheng et al., 2014; Sharan et al., 2024; Yang et al., 2024) construct a finite automaton representing video frame sequences and verify this automaton against temporal logical specifications. However, their approaches do not account for uncertainties related to the vision-based detection algorithms (Bhatt et al., 2024). Moreover, the construction and verification of automatons cannot be done in real-time.

**Mathematical Approaches for Privacy Preservation**   A parallel line of works use differential privacy or cryptography to provide theoretical guarantees without deep learning models. For instance, Cangialosi et al. (2022) developed a differential privacy mechanism to protect video privacy, and Rahman et al. (2012) proposed a cryptographic approach for video privacy. However, such methods require an exhaustive description of all privacy specifications beforehand, and thus do not generalize well for real-world applications, where previously undefined privacy-sensitive objects may emerge in the future. Such a capability cannot occur without some degree of integration with deep learning models to allow open-set detection. In contrast, our method enforces the video to satisfy any complex privacy requirements expressed in logic formulas by developing efficient formal method abstraction techniques capable of real-time deployment and integrating them with off-the-shelf deep learning models for enhanced generalization capabilities.

## 3 Problem Formulation

A video $\mathcal{V}$ is a sequence of frames $\mathcal{F}_1, ..., \mathcal{F}_k$ where each $\mathcal{F}_k \in \mathbb{R}^{C \times W \times H}$ is an RGB image with $C$ channels, $W$ width, and $H$ height. A video can be prerecorded or live-streamed from sources such as autonomous vehicles or security cameras.

We define a **privacy specification** $\Phi$ as a temporal logic formula (Rescher & Urquhart, 2012) constraining the appearance of privacy-sensitive objects. Since we want to preserve privacy at all times, we express a privacy specification as $\Phi = \Box(\tilde{\Phi})$, where $\Box$ represents the "ALWAYS" temporal operation and $\tilde{\Phi}$ is a first-order logic formula (Barwise, 1977). The presence of privacy-sensitive objects is constrained by $\Phi$.

We define a set of atomic propositions $AP$, where each proposition $p_i \in AP$ is a textual description of a privacy-sensitive object. Then, we use an object detection model (ODM), $\mathcal{M}_{obj}$, to detect these objects. $\mathcal{M}_{obj} : \mathbb{R}^{C \times W \times H} \times AP \to [0, 1]$ takes a frame $\mathcal{F}_k \in \mathbb{R}^{C \times W \times H}$ and a proposition $p_i \in AP$ as inputs, and returns a confidence score $c \in [0, 1]$, denoted as $c = \mathcal{M}_{obj}(\mathcal{F}_k, p_i)$. However, deep learning models are often overconfident, and their detection accuracy cannot be guaranteed. Therefore, we calibrate the confidence using conformal prediction (Shafer & Vovk, 2008), which provides a lower bound for the probability of correctly detecting privacy-sensitive objects in every frame, considering the inherent uncertainty in deep learning model predictions.

However, traditional conformal prediction approaches focus on post-processing and do not account for temporal events. Therefore, we use calibrated confidence to detect and constrain privacy-sensitive objects over time and provide a probabilistic guarantee on a sequence of frames.

To achieve this, we develop an algorithm $\mathcal{VA}$ that takes a sequence of $k$ frames and returns a formally verifiable video abstraction $\mathcal{A}_k$ encoding the object detection across the sequence: $\mathcal{VA}([\mathcal{F}_1, ..., \mathcal{F}_k]) = \mathcal{A}_k$. The **video abstraction** $\mathcal{A}_k$ is represented as a labeled Markov chain, detailed rigorously in Section 4 as

it requires extensive background and mathematical notation. This provides a probabilistic guarantee on a frame sequence via formal verification (Woodcock et al., 2009).

**Definition 1** (**Probabilistic Guarantee on a Frame Sequence**). Given a sequence of frames $\mathcal{F}_1, ..., \mathcal{F}_k$, a privacy specification $\Phi$, and a video abstraction $\mathcal{A}_k$ at the $k^{\text{th}}$ frame, a probabilistic guarantee $\mathcal{PG}_k(\mathcal{A}_k \models \Phi)$ on the frame sequence $\mathcal{F}_1, ..., \mathcal{F}_k$ represents the theoretical minimum probability that the presence of privacy-sensitive objects in the frame $\mathcal{F}_1$ through $\mathcal{F}_k$ adheres to $\Phi$.

**Problem 1** (**Real-Time Video Privacy Preservation**). Given a frame sequence $\mathcal{F}_1, ..., \mathcal{F}_{k-1}$, an incoming frame $\mathcal{F}_k$ from a video stream, a privacy specification $\Phi$, and an algorithm $\mathcal{VA}$ that builds a video abstraction from the frame sequence, we aim to remove privacy-sensitive objects in $\mathcal{F}_k$ such that $\mathcal{A}_k = \mathcal{VA}([\mathcal{F}_1, ..., \mathcal{F}_k])$ satisfies $\Phi$ with a probability at least $\mathcal{PG}_k(\mathcal{A}_k \models \Phi)$.

## 4 Privacy-Constrained Video Streaming

We develop privacy-constrained video streaming (`PCVS`), a method to enforce live video streams that satisfy a user-given privacy specification with a probabilistic guarantee. The overall pipeline for `PCVS` is illustrated in fig. 1.

**Real-Time Video Privacy Preservation Framework** We explain our framework with a running example in a real-time video stream from a real robot (see Figure 2). We aim to hide human faces so that no personal identity will be revealed in vision-based robot operations. Therefore, the privacy specification is $\Phi = \Box \, \text{person} \rightarrow \neg \text{face}$, where $\rightarrow$ and $\neg$ mean "implies" and "not", respectively. We detect humans and faces at every frame via the ODM. Subsequently, we use conformal prediction to obtain a calibrated confidence score for the detection in the current frame. We build a video abstraction $\mathcal{A}_k$ to represent the detection results for humans and faces across a sequence of frames and utilize it to obtain a probabilistic guarantee on $\Phi$ being satisfied. We then verify if the guarantee is above the user-given privacy threshold $\lambda \in [0, 1]$. If this threshold is not met, we iteratively remove the detected faces and update the guarantee $\mathcal{PG}_k(\mathcal{A}_k \models \Phi)$ until the threshold is met.

### 4.1 Probabilistic Guarantee on Video Privacy

Given a sequence of $k$ frames and a privacy specification $\Phi$, we compute a probabilistic guarantee $\mathcal{PG}_k(\mathcal{A}_k \models \Phi)$. This guarantee is updated at each incoming frame. We use formal methods to prove that the guarantee holds.

**Confidence Calibration via Conformal Prediction** Recall that an ODM $\mathcal{M}_{obj}(x_i, y_i) = c$ receives an image $x_i$ and a textual object label $y_i$ as a prompt and returns a confidence score $c \in [0, 1]$. Given the ODM $\mathcal{M}_{obj}$ and a labeled calibration dataset that is distributed identically to the task domain, using conformal prediction (Shafer & Vovk, 2008), we learn a calibration function $f_C : [0, 1] \mapsto [0, 1]$ that maps a confidence score, $c \in [0, 1]$ to a lower bound for the probability of correct detection.

We first collect a calibration set $\{(x_i, y_i)\}_{i=1}^m$ consisting of $m$ (image, ground truth text label) tuples. Then, we apply $\mathcal{M}_{obj}$ to detect the privacy-sensitive objects in the images $\{x_i\}_{i=1}^m$ and get a set of *non-conformity scores*: $\{1 - \mathcal{M}_{obj}(x_i, y_i)\}_{i=1}^m$. A non-conformity score is the sum of confidence scores of wrong detections. Next, we estimate a probability density function of these scores, denoted as $f_{nc}(z)$,

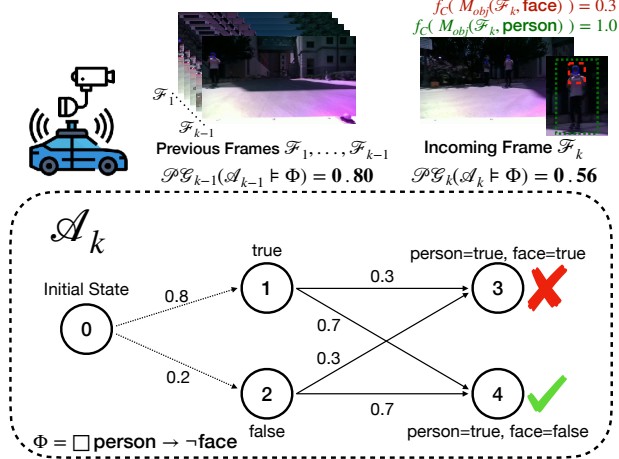

Figure 2: A running example on how to compute the probabilistic guarantee via video abstraction.

where $z$ is a nonconformity score. Then, we use Theorem 1 to establish a theoretical lower bound for the probability of the correct detection. The objective of confidence calibration is specifically to establish a theoretical bound on the privacy-preservation success ratio, rather than directly enhancing performance. This calibration provides rigorous privacy assessments in real-time streaming scenarios, especially when ground truth data is unavailable to the user.

**Theorem 1** (Shafer & Vovk (2008), Section 4). Let $\epsilon \in [0, 1]$ be a pre-defined error bound and $x_n$ be an image outside the calibration set. We define a *prediction band* as $\hat{C}(x_n) = \{p_i : \mathcal{M}_{obj}(x_n, p_i) \geq 1 - c^*, p_i \in AP\}$. Then, according to conformal prediction, there exists a confidence $c^*$ such that $\epsilon = 1 - \int_0^{c^*} f_{nc}(z)dz$ satisfies $\mathbb{P}\left[y_n \in \hat{C}(x_n)\right] \geq 1 - \epsilon$, where $y_n$ is the ground truth label for $x_n$. The proof is in Section 4, Shafer & Vovk (2008).

Note that $\mathcal{M}_{obj}(x_i, p_i)$ returns a single confidence score indicating whether $p_i$ exists in $x_i$. By the theory of conformal prediction, $1 - \epsilon$ is a theoretical lower bound for the probability of the ground truth label belonging to the prediction band $\hat{C}(x_n)$. If $\mathcal{M}_{obj}(x_i, p_i) > 0.5$, we provide a lower bound for the probability of the existence of $p_i$. Otherwise, if $\mathcal{M}_{obj}(x_i, p_i) \leq 0.5$, we bound the probability of non-existence. Hence, we get a calibration function

$$f_C(c) = \begin{cases} \int_0^c f_{nc}(z)dz, & \text{if } c > 0.5 \\ \int_0^{1-c} f_{nc}(z)dz, & \text{otherwise.} \end{cases} \tag{1}$$

**Video Abstraction**   For verifying a *real-time* video stream against the privacy specification $\Phi$, a key challenge is to verify the temporal behaviors of *all* the previously received frames plus the current frame. This makes verification space- and time-inefficient because we must repeatedly verify previous frames for each new incoming frame. To overcome this challenge, we build an abstraction for the video stream, which enables real-time verification.

**Definition 2** (**Video Abstraction**). A video abstraction is a labeled Markov chain $(S, s_0, P, L)$, where $S$ is a set of states, each state corresponds to a conjunction of atomic propositions, $s_0 \in S$ is the initial state, $P : S \times S \to [0, 1]$ is a transition function. $P(s, s')$ represents the probability of transition from a state $s$ to a state $s'$ and $\sum_{s' \in S} P(s, s') = 1$. $L : S \to 2^{AP}$ is a labelling function.

---

Algorithm 1: Real-Time Video Abstraction

**Require:** object detection model $\mathcal{M}_{obj}$, calibration function $f_C$,
  propositions set $AP$, specification $\Phi$, probability $p_{k-1}$
  of previous frames satisfying $\Phi$, incoming frame $\mathcal{F}_k$
1: $S_{\text{obs}}, P, L = \{\}, \{\}, \{\}$                                     ▷ Initialize the abstraction
2: $S_{\text{obs}}.\text{add}(0), S_{\text{obs}}.\text{add}(1), S_{\text{obs}}.\text{add}(2)$          ▷ We represent each state with an Arabic numeral
3: $L(1) = \text{false}, L(2) = \text{true}$
4: $P(0, 1) = 1 - p_{k-1}, P(0, 2) = p_{k-1}$      ▷ Add transitions to indicate the probability of previous frames
  satisfying $\Phi$
5: $i = 3$                                              ▷ Initialize a indexer representing states
6: **for** $\sigma$ in $2^{AP}$ **do**                              ▷ $\sigma$ is a conjunction of atomic propositions
7:     $\text{prob} = \prod_{p \in \sigma} f_C(\mathcal{M}_{obj}(\mathcal{F}_k, p))$                   ▷ Get a lower bound for a detection result
8:     **if** $\text{prob} > 0$ **then**                   ▷ Add a state to represent the detection with the lower bound
9:         $S_{\text{obs}}.\text{add}(i), L(i) = \sigma, P(1, i) = \text{prob}, P(2, i) = \text{prob}, i = i + 1$
10:    **end if**
11: **end forreturn** $S_{\text{obs}}, s_0 = 0, P, L$

---

We propose Algorithm 1 to build video abstractions. We demonstrate it through an example in fig. 2. First, we add an initial state (State 0 in fig. 2), a state representing the event that all previous frames (if they exist) satisfy $\Phi$ (State 1), and a state representing the event that previous frames fail $\Phi$ (State 2), as in lines 1-3. Next, we add transitions from State 0 to State 1 and to State 2 with the probability of previous frames satisfying $\Phi$ as in line 4. Then, we detect objects in the incoming frame $\mathcal{F}_k$ and get the probability bound

for correct detection. For each conjunction of propositions (e.g., person=true and face=false), we build a state and add transitions to this state with the probability bound of correctly detecting objects described in this conjunction, as in lines 6-9. Hence, we obtain the video abstraction $\mathcal{A}_k$ (e.g., fig. 2).

Following Algorithm 1, we incrementally add new states to the abstraction (rather than build a new one) with the arrival of each new incoming frame and check it against $\Phi$. Hence, this abstraction can be used to check video streams efficiently. Then, we theoretically prove that the probabilistic guarantee obtained through this abstraction holds.

**Probabilistic Guarantees for Frame Sequence**   Given a video abstraction $\mathcal{A} = (S, s_0, P, L)$, we define a *path* $\pi$ as a sequence of states starting from $s_0$. The states evolve according to the transition function $P$. A *prefix* is a finite path fragment starting from $s_0$. We define a *trace* as $\psi = \text{trace}(\pi) = L(s_0)L(s_1)L(s_2)\ldots$, where $s_0, s_1, s_2, \ldots \in \pi$. $\text{Traces}(\mathcal{A})$ denotes the set of all traces from $\mathcal{A}$. Each trace $\psi = L(s_0)L(s_1)L(s_2)\ldots$ is associated with a probability $\mathbb{P}(\psi) = P(s_0, s_1) \times P(s_1, s_2) \times \ldots$

The privacy specification is in the form of $\square \tilde{\Phi}$. Hence, a privacy specification describes a *safety property* (Baier & Katoen, 2008).

**Definition 3** (**Safety Property**). A safety property $P_{\text{safe}}$ is a set of traces in $(2^{AP})^\omega$ ($\omega$ indicates infinite repetitions) such that for all traces $\psi \in (2^{AP})^\omega \backslash P_{\text{safe}}$, there exists a finite prefix $\hat{\psi}$ of $\psi$ such that $P_{\text{safe}} \cap \{\psi' \in (2^{AP})^\omega \,|\, \hat{\psi} \text{ is a prefix of } \psi'\} = \Phi$. $\hat{\psi}$ is a *bad prefix* and $\text{BadPref}(P_{\text{safe}})$ is the set of all bad prefixes with respect to $P_{\text{safe}}$.

A video satisfies the privacy specification if its abstract representation $\mathcal{A}$ satisfies the safety property $P_{\text{safe}}$, i.e., $\text{Traces}(\mathcal{A}) \subseteq P_{\text{safe}}$. The probability that a video satisfies the specification is

$$\mathbb{P}[\mathcal{A} \text{ is safe}] = \mathbb{P}[\pi \in \text{path}(s_0) \,|\, \text{trace}(\pi) \in P_{\text{safe}}] = \sum_{\psi \in \text{Traces}(\mathcal{A}) \cap P_{\text{safe}}} \mathbb{P}(\psi). \tag{2}$$

Note that this probability is a probabilistic guarantee on a sequence of frames. According to the definition of safety property, we derive the following theorem:

**Theorem 2.** Consider a set of prefixes $\hat{\Psi}$ such that $\mathbb{P}\{\hat{\psi} \in \hat{\Psi} \,|\, \hat{\psi} \notin \text{BadPref}(P_{\text{safe}})\} \geq \alpha$. Let $\bar{S} \subset S$ be a subset of states in $\mathcal{A}$ such that $\mathbb{P}\{\hat{\psi}L(s) \notin \text{BadPref}(P_{\text{safe}}) \,|\, \hat{\psi} \notin \text{BadPref}(P_{\text{safe}}) \text{ and } s \in \bar{S}\} \geq \beta$. Then, $\mathbb{P}\{\hat{\psi}L(s) \notin \text{BadPref}(P_{\text{safe}}) \,|\, s \in \bar{S} \text{ and } \hat{\psi} \in \hat{\Psi}\} \geq \alpha\beta$.

*Proof.* Let $A = \{\hat{\psi}L(s) \notin \text{BadPref}(P_{\text{safe}}) \,|\, s \in \bar{S} \cap \hat{\psi} \in \hat{\Psi}\}$ and $B = \{\hat{\psi} \notin \text{BadPref}(P_{\text{safe}}) \,|\, \hat{\psi} \in \hat{\Psi}\}$. Then, $A|B = \{\hat{\psi}L(s) \notin \text{BadPref}(P_{\text{safe}}) \,|\, s \in \bar{S} \cap \hat{\psi}L(s) \notin \text{BadPref}(P_{\text{safe}})\}$ and $\mathbb{P}(A) = \mathbb{P}(A|B) \cdot \mathbb{P}(B) \geq \alpha\beta$.  $\square$

From Theorem 2, we can compute a new probabilistic guarantee on a sequence of frames after each incoming frame. However, the length of the abstraction's prefixes increases as the stream continues, leading to high complexity. Therefore, we derive the following proposition to show that Theorem 2 holds even if we fix the length of the prefixes (proof of the proposition is in the Appendix):

**Proposition 1.** Let $\hat{\psi}_T$ and $\hat{\psi}_F$ be single element prefixes whose corresponding paths only consist of one state such that $\hat{\psi}_T \notin \text{BadPref}(P_{\text{safe}})$ and $\hat{\psi}_F \in \text{BadPref}(P_{\text{safe}})$. Let $\mathbb{P}(\hat{\psi}_T) = \alpha$, $\mathbb{P}(\hat{\psi}_F) = 1 - \alpha$, $\Psi' = \{\psi_T, \psi_F\}$, then if we replace $\hat{\Psi}$ with $\Psi'$ in Theorem 2, the Theorem still holds.

*Proof.* Since $\mathbb{P}(\hat{\psi}_T) = \alpha$ and $\mathbb{P}(\hat{\psi}_F) = 1 - \alpha$, $\mathbb{P}\{\hat{\psi} \in \Psi' | \hat{\psi} \notin \text{BadPref}(P_{\text{safe}})\} = \alpha$. The replacement of $\hat{\Psi}$ with $\Psi'$ does not affect $\mathbb{P}\{\hat{\psi}L(s) \notin \text{BadPref}(P_{\text{safe}}) \,|\, \hat{\psi} \notin \text{BadPref}(P_{\text{safe}}) \cap s \in \bar{S}\}$. Thus, the conditions of Theorem 2 are satisfied, and $\mathbb{P}\{\hat{\psi}L(s) \notin \text{BadPref}(P_{\text{safe}}) \,|\, s \in \bar{S} \cap \hat{\psi} \in \Psi'\} \geq \alpha\beta$.  $\square$

From Theorem 2 and Proposition 1, we can compute $\mathcal{PG}_k(\mathcal{A}_k \models \Phi)$ as follows:

$$\mathcal{PG}_k(\mathcal{A}_k \models \Phi) = \mathcal{PG}_{k-1}(\mathcal{A}_{k-1} \models \Phi) \times \left( \sum_{\sigma \models \tilde{\Phi}} \prod_{p \in \sigma} f_C(\mathcal{M}_{obj}(\mathcal{F}_k, p)) \right) \tag{3}$$

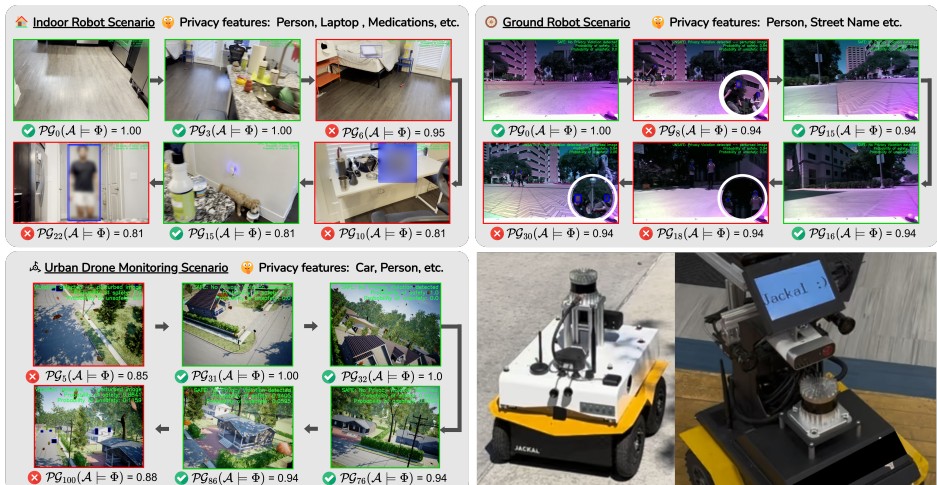

Figure 3: We present demonstrations for indoor robot navigation (top left), ground robot navigation (top right), and urban drone monitoring (bottom left). The indoor and ground robots are shown in the bottom right. Scenes with an 'x' in a red circle contain privacy-sensitive objects, and our method successfully conceals them. All demonstrations effectively maintain privacy above the user-given privacy threshold of 0.80, denoted as $\mathcal{PG}_k(\Phi) > 0.80$.

The video abstraction captures all previous frames in only two states (States 1 and 2 in fig. 2) instead of accumulating states for every frame in the sequence. Thus, we can efficiently update the guarantee through a single computation. In fig. 2, $\mathcal{PG}_k(\mathcal{A}_k \models \Phi) = 0.8 \times 0.7 = \mathbf{0.56}$.

## 5 Robot Demonstrations

Our experiments assess our method in two areas: (i) its ability to protect privacy, and (ii) its efficiency in preserving other features for vision-based robot tasks.

We demonstrate our approach on a Jackal ground robot for autonomous driving, an indoor robot for in-house navigation and service, and a drone for urban monitoring (see Figure 3). Given video streams from robot cameras, we aim to execute actions based on the control policy ($\pi$). Our approach effectively preserves privacy with formal guarantees without compromising performance in the real-time robot operation. We use YOLOv9 (Wang et al., 2025) in our method for all demonstrations.

**Indoor Navigation** In the first demonstration, we deploy PCVS to an indoor navigation robot to protect user privacy. We ground the robot in a private residence for in-house services such as transporting objects and house cleaning. While the robot perceives the environment through visual observations, we aim to preserve the privacy within such observations. The privacy specification is $\Phi_1 = \Box \, (\neg \, \text{laptop} \wedge \neg \, \text{medication} \wedge \neg \, \text{person})$, which requires hiding all people, laptops, and medications appearing in the scenes. Figure 3 (top left) demonstrates how our method performs to conceal the sensitive objects such that the video satisfies the privacy specification. The probabilistic guarantee of privacy preservation throughout the complete operation is 0.81.

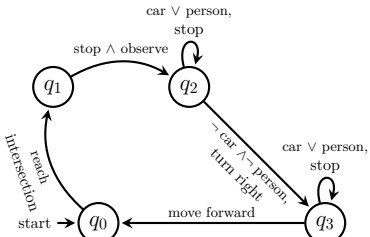

Figure 4: A sample control policy for the ground robot. Each transition is associated with an (input, output) tuple.

**Ground Robot Driving** We deploy the control policy on the ground robot for five driving tasks, such as turning right at a stop sign (as presented in fig. 4). We embed PCVS in the robot's camera to conceal

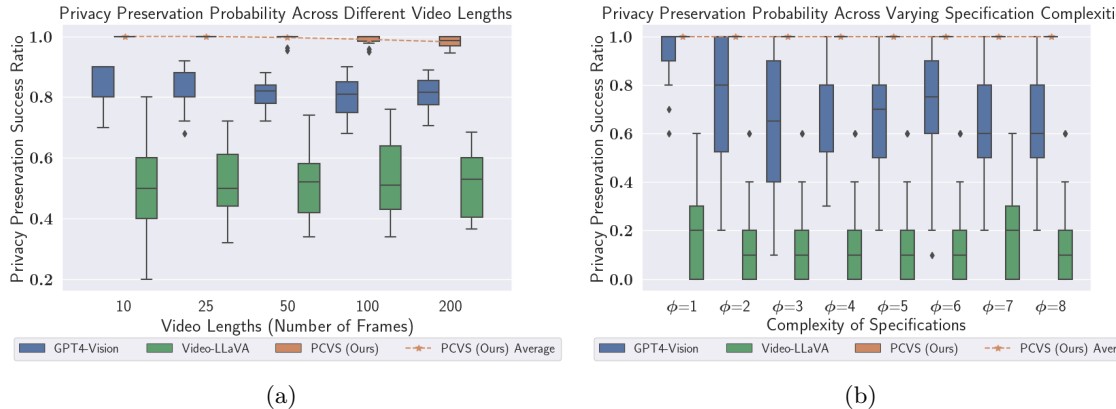

Figure 5: **PCVS effectively maintains privacy in long-horizon videos and complex privacy specifications**. In fig. 5a, PCVS consistently preserves privacy, achieving an average Privacy Preservation Success Ratio of 0.97 across various video lengths. In fig. 5b, we show that PCVS consistently upholds privacy regardless of the complexity of specifications with an average Privacy Preservation Success Ratio of 0.94.

sensitive objects during real-time operation. In the driving example, we define a set of privacy specifications:

$$\Phi_2 = \Box \, \neg \, \text{road sign},$$
$$\Phi_3 = \Box \, ((\text{bicycle} \rightarrow \neg \, \text{person}) \vee (\text{person} \rightarrow \neg \, \text{face})),$$
$$\Phi_4 = \Box \, ((\text{bus} \vee \text{car}) \rightarrow \neg \, \text{plate}).$$

Intuitively, we want to conceal privacy-sensitive objects such as road name signs, car plates, and human faces. Figure 3 (top right) presents an example of how our method conceals sensitive objects to satisfy the specifications. The driving operation has a probabilistic guarantee of privacy preservation at 0.84, i.e., at least 84 percent of satisfying the specifications

When concealing sensitive objects, it is crucial to ensure that such concealment does not adversely impact the robot's decision-making processes. For instance, the robot should still be capable of detecting and avoiding pedestrians even after their faces have been obscured. More precisely, consider a safety specification $\Phi_5 = \Box \, ((\text{car} \vee \text{person}) \rightarrow \mathbf{X} \, \text{stop},$ which necessitates that the robot comes to a stop if cars or pedestrians are present ahead. In the demonstration, the robot performs identically regardless of whether our method is deployed, and in both cases, it satisfies the safety specifications. Hence, we show that our privacy

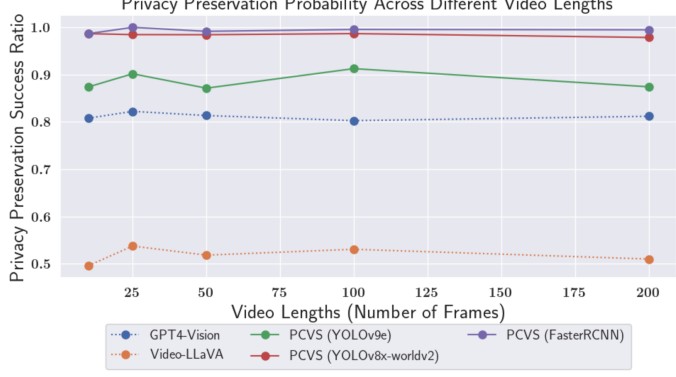

Figure 6: The comparisons between our method and other benchmarks and the comparisons between our method under different object detection models.

protection will not over-conceal non-sensitive objects and negatively impact the decision-making procedure in driving scenarios.

**Urban Drone Monitoring**   We demonstrate the applicability and effectiveness of our method in urban drone monitoring scenarios. The privacy specification is $\Phi_6 = \Box \, ((\text{bicycle} \rightarrow \neg\text{person}) \wedge (\text{car} \vee \text{bus} \rightarrow \neg \, \text{person}))$, which requires hiding all cars, buses, bicycles, and persons appearing in the scenes. Figure 3 (bottom left) demonstrates the successful performance of our method in concealing objects that are irrelevant to the monitoring task, yet require privacy preservation. The demonstration also indicates the real-time capability of our method to be seamlessly migrated to real-world drone applications.

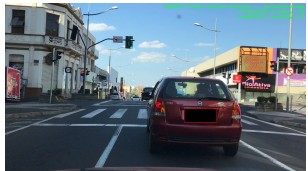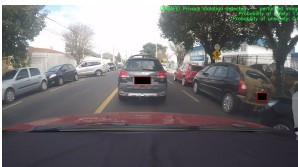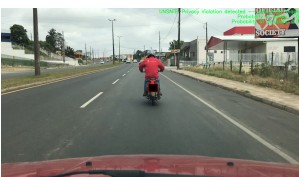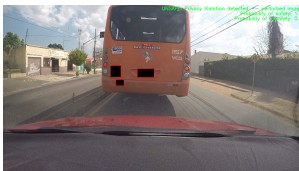

Figure 7: Demonstrations on our method concealing license plates in the driving scenes from the UFPR-ALPR dataset.

## 6 Quantitative Analyses

We present quantitative analyses in two areas: preserving privacy and preserving non-private visual features. We use YOLOv9 (Wang et al., 2025) on large-scale image datasets—ImageNet (Deng et al., 2009) and MS COCO (Lin et al., 2014)—and a real-world driving dataset—UFPR-ALPR (Laroca et al., 2018). We choose foundation models over conventional object detectors such as Faster-RCNN due to their inherent ability to handle temporal reasoning across frame sequences, essential for robust privacy-preservation in streaming applications.

Our analyses show `PCVS` can preserve privacy even for long-horizon videos with complex privacy specifications. We define the **complexity of specifications**, $\phi$, as the *number of propositions* in $\Phi$. For instance, the complexity of a specification $\Phi = \Box(\neg p_1 \wedge \neg p_2 \vee \neg p_3)$ with propositions $AP = \{p_1, p_2, p_3\}$ is $\phi = 3$.

*Evaluation Dataset I (ED 1)*: We focus on the presence of "person" in videos. We select images of a person from the ImageNet dataset and randomly insert these images at various positions for each video duration, filling any remaining slots with random images. We produce five different video lengths: 10, 25, 50, 100, and 200, with 25 video samples for each duration, resulting in 125 video samples overall.

*Evaluation Dataset II (ED 2)*: We use the MS COCO dataset to evaluate our method at different complexities of specifications because it has multiple labels per image. Each level of specification complexity consists of 20 video samples, with an average of 50 frames per sample, resulting in a total of 160 videos. This dataset was developed using the same process as the ED1 dataset, with modifications made to accommodate the complexity of the specifications. For example, if $\phi = 3$, the privacy specification for the dataset is $\Phi = \neg p_1 \wedge \neg p_2 \wedge \neg p_3$, where $p_1$, $p_2$, and $p_3$ are the ground truth labels of the selected image. These images are then randomly placed within the dataset, and the remaining slots are filled with random images.

*Evaluation Dataset III (ED 3)*: We randomly select a subset of images from the UFPR-ALPR dataset (Laroca et al., 2018) to generate videos with lengths 10, 25, 50, 100, and 200. Each length has 200 video samples (a total of 1000 videos). The dataset consists of labeled images that include driving-related objects such as vehicles, license plates, etc. We form a video by integrating a sequence of images from the dataset.

*Benchmarks*: To assess the ability of `PCVS` to detect privacy violations based on a given privacy specification, we use GPT-4 Vision (OpenAI, 2023) and Video LLaVA (Lin et al., 2023) as benchmarks. This is because both benchmark methods can process a sequence of images from a video alongside a privacy specification.

### 6.1 Privacy Preservation

We quantitatively evaluate the performance of privacy preservation across varying video lengths and specification complexities. For this evaluation, we define a metric representing the success ratio of privacy preservation as follows:

$$\text{Privacy Preservation Success Ratio} = \frac{\text{Number of } p_i \in AP \text{ detected or concealed}}{\text{Total number of private } p_i \in AP \text{ within } \mathcal{V}}.$$

`PCVS` counts the number of concealed objects, while the benchmark counts the number of detected objects, as the benchmark does not natively conceal those objects. We assume that if those objects are detected, they can be concealed by other methods, such as Gaussian concealing. During the concealment process, the user-defined privacy specifications determine the privacy-sensitive objects to be concealed. For instance,

in the paper, $\Phi_1 - \Phi_4$, define the combination of privacy-sensitive objects to be concealed. It is important to note that in our experiments, all privacy specifications are prioritized equally, and the propositions are defined so that non-private visual features are unaffected.

**Comparison by video length** Our method ensures privacy preservation in live video streams, which means that the video length can be infinite. Hence, it is crucial to assess whether privacy is maintained as video streams become longer. To this end, we test `PCVS` on videos with various lengths from ED 1. We find that `PCVS` consistently maintains performance in preserving privacy, in contrast to benchmark methods that exhibit degraded performance as the length of videos increases (see fig. 5a).

We also examine how the underlying vision model ability affects the privacy preservation ratio. We repeat the experiment on ED 1 while using different detection models: Yolov9e (Wang et al., 2025), Yolov8x-worldv2 (Cheng et al., 2024), and Faster-RCNN (Ren et al., 2016), all with default parameters. We present the privacy preservation success ratio of our method using different detection models

Figure 8: **Effective privacy preservation in real-world driving scenes.** We apply `PCVS` to driving scenes from the UFPR-ALPR dataset and conceal privacy-sensitive objects such as license plates. The privacy preservation success ratio of `PCVS` is consistently above 0.8.

versus video lengths in fig. 6. The results show that our method is sensitive to the detection model quality. Our method outperforms the benchmarks (GPT4-Vision and Video-LLaVA) at every video length when using the mainstream detection models.

To further demonstrate the real-world applicability of our method, we apply it to ED 3 and evaluate the privacy preservation success ratio across different video lengths. Recall that ED 3 consists of images collected from vehicle dash cameras. Figure 8 shows our method's high privacy preservation success ratio—consistently above 0.8 regardless of length. We present some demonstration figures in Figure 7. The results indicate the applicability of our method to real-world tasks such as autonomous driving.

**Comparison by specification complexities** Next, we assess `PCVS` based on the complexity of specifications. This comparison is important because a privacy specification can be intricate, involving more than just two or three propositions. For example, a specification might require the detection and concealment of multiple privacy-sensitive objects within the same video, such as faces, license plates, and specific types of clothing. Our method significantly outperforms benchmark methods (see Figure 5b) regardless of the complexity of specifications. We demonstrate that `PCVS` can effectively handle highly complex privacy compositions in real-time video streams, ensuring robust privacy protection.

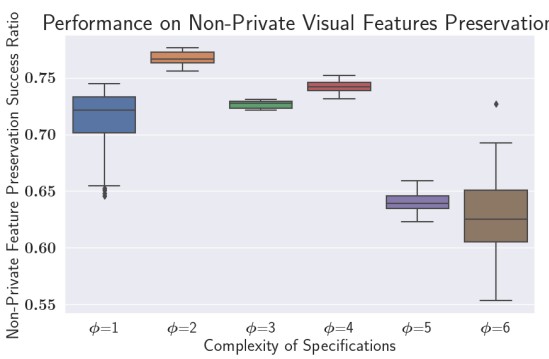

Figure 9: **Preserving non-private visual features for vision-based robot operation.** Our method can detect non-private objects after concealing private objects specified in $\Phi$. However, performance degrades from $\phi = 5$ because more private objects get concealed indirectly masking out other objects.

## 6.2 Non-private Visual Feature Preservation

Preserving non-private visual features is crucial for vision-based robot operation, as it relies on visual observation for control policies. In our demonstration (as presented in fig. 3), the ground robot must be capable of identifying people from privacy-constrained video footage to make appropriate decisions, such as stopping.

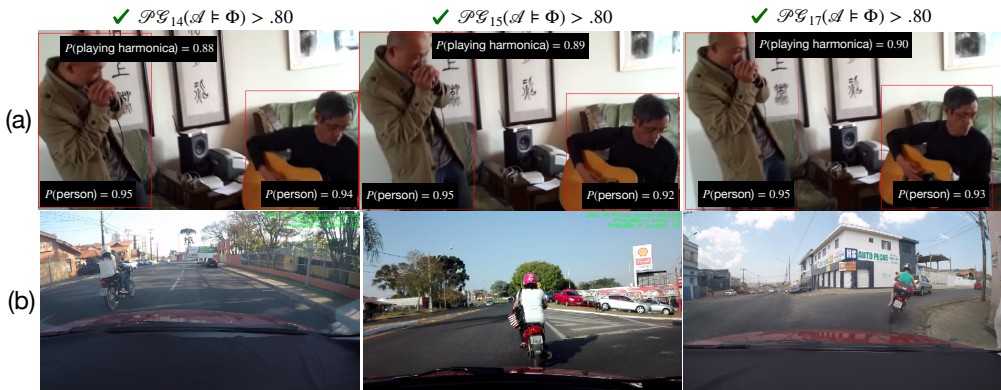

Figure 10: **(a)** result for $\square$(person $\wedge$ playing harmonica); the object and activity are not concealed for demonstration purposes. **(b)** failure cases from the UFPR-ALPR dataset.

We analyze how our method preserves non-private visual features using ED2 in fig. 9. We define a metric that represents the success ratio of preserving non-private visual features as follows:

$$\text{Non-Private Feature Preservation Success Ratio} = \frac{\text{Number of } \chi \text{ detected after concealing } p_i \in AP}{\text{Total number of } \chi \text{ within } \mathcal{V}},$$

where $\chi$ is a non-private target object for detection. In our evaluation, non-private visual features remain preserved and detectable even after the concealment of privacy-sensitive objects as defined in the privacy specifications. However, the success ratio of non-private preservation decreases as the complexity of these specifications increases. This is because `PCVS` conceals a larger image area as the number of privacy-sensitive objects increases.

## 6.3 Computational Complexity

Model checking often incurs significant computational overhead, particularly as the state space size increases with the number of video frames, which limits its capability in real-time applications. Hence, we develop an abstraction method to resolve this limitation.

Figure 11 shows the verification time versus the frame number under different numbers of atomic propositions. The experiments are performed using the video collected by the ground robot with an *Apple M2 CPU*. As the frame number increases, the verification time without our video abstraction method grows linearly, while the time with abstraction remains constant.

Furthermore, we tested our method with YOLOv9 on both *Intel Xeon Gold* CPU and *Nvidia A5000* GPU. The average runtimes for processing one proposition in one image frame (1600×900 pixels) using CPU and GPU are

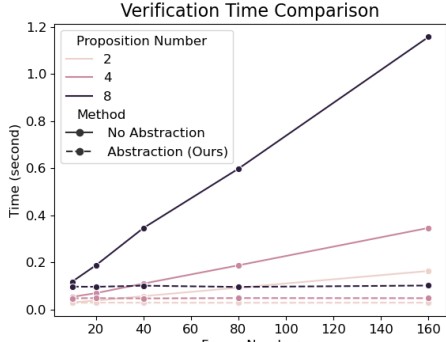

Figure 11: Comparison between the verification time with and without model abstraction. Our model abstraction significantly reduces the latency and maintains a constant latency when the proposition number increases.

159 milliseconds and 69 milliseconds. Therefore, a robot with a CPU can also preserve privacy in real-time at an approximate six frames per second (fps) frequency, and a robot with a GPU is capable of videos with 14 fps. We present more details in fig. 12.

## 6.4 Ablation Studies

**Action-Based Privacy Constraints** We deploy a Video MAE activity recognition model replacing the object detection model to show the capability of our framework in detecting action-based privacy (See fig. 10

(a)). The experiment demonstrates how privacy considerations can go beyond static object detection and account for dynamic behaviors or activities that may be sensitive, such as gestures or interactions that reveal identity, intent, or context. Integrating an activity recognition model allows the proposed method to support fine-grained control over temporal privacy, healthcare, or smart home environments where actions can be private even when identities are anonymized.

**Failure Cases & Stress Testing** We conduct an evaluation of failure cases and stress test the system to assess its robustness. The 5 percent failure cases are marginal errors inherent to the neural network (i.e., object detection models). An additional experiment in fig. 10 (b) shows that most failure cases involve motorcycle license plates due to the poor performance of the detection model. This underscores the importance of domain-specific training, particularly for small or occluded objects that differ significantly in shape or layout from more commonly detected categories (e.g., car plates). These failures highlight potential areas for improving detection robustness, such as augmenting the dataset with diverse examples or integrating ensemble models. Moreover, stress testing reveals that the system maintains reliability under typical variance but is susceptible to degradation in edge scenarios—an important consideration for real-world deployment.

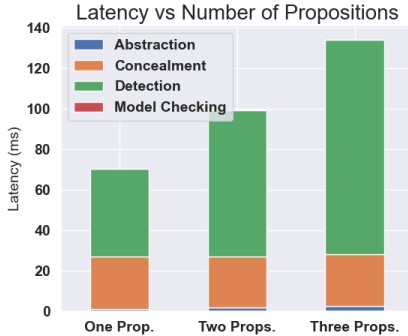

Figure 12: Latency comparison in each stage of our method to process one frame. Object detection takes up the majority of the time, and the time required grows linearly as the number of propositions increases. The time for other stages is negligible or remains constant across different numbers of propositions.

## 7 Conclusion

**Summary** We propose `PCVS`, a novel method for enforcing privacy in live video streams, particularly those generated by robotic systems or consumed by robot learning algorithms. `PCVS` provides a probabilistic guarantee that user-defined privacy specifications are satisfied, offering a principled and flexible approach to privacy preservation in dynamic environments. Our method significantly outperforms state-of-the-art baselines on both short and long video sequences regarding accuracy, efficiency, and adaptability. In addition, we demonstrate the real-time performance and practical utility of `PCVS` across three distinct robotic applications, highlighting its robustness and applicability in real-world scenarios such as autonomous navigation, human-robot interaction, and manipulation tasks. These results establish `PCVS` as a promising solution for privacy-aware robotics and embodied AI.

**Limitations and Future Work** While `PCVS` demonstrates strong performance and real-time capability, it remains limited by the representational and reasoning capacity of the underlying object detection model. This affects the range and precision of privacy specifications that can be effectively interpreted and enforced, especially in cases requiring understanding context or abstract concepts. Another limitation is the reliance on pre-defined privacy specifications, which may not fully capture evolving or user-specific privacy concerns in dynamic environments. As a future direction, we aim to extend our framework to support more expressive and generalized privacy specifications, moving beyond safety properties to include formal notions such as liveness (ensuring events eventually occur) and fairness (ensuring unbiased treatment across individuals or groups). Such extensions would improve the flexibility, interpretability, and applicability of `PCVS` in broader autonomous and interactive system settings.

**Broader Impact Statement**

This work introduces a method to enforce privacy constraints in real-time video streams from robot cameras, ensuring sensitive objects are concealed based on predefined logical specifications. Unlike existing approaches, it offers theoretical guarantees and works in live deployments without degrading robots' visual performance. By enabling privacy-aware AI systems, this method facilitates the ethical use of robotics in privacy-sensitive environments, such as homes, hospitals, and workplaces.

**Acknowledgments**

This material is based upon work supported in part by the Office of Naval Research (ONR) under Grant No. N00014-22-1-2254 and No. N00014-21-1-2502. Additionally, this work was supported by the Defense Advanced Research Projects Agency (DARPA) contract DARPA ANSR: RTX CW2231110 and the National Science Foundation under Grant No. 2409535. Approved for Public Release, Distribution Unlimited.

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

## A  Additional Details on Framework Implementation

**Calibration**  In our PCVS framework, we construct a global calibration set across all object labels and build a single non-conformity distribution shared by all detections. This design supports open-vocabulary privacy constraints, where sensitive objects may not belong to a fixed label set. While class-conditional calibration can yield tighter bounds for closed vocabularies, global calibration offers better generalizability, at the cost of a trade-off.

**Composition of Multiple Propositions**  We apply conjunctions when multiple privacy-sensitive objects are involved. Our framework is designed to provide lower-bound guarantees, prioritizing conservative estimates to maintain safety and reliability.

**Threat Model and Complete Concealment**  PCVS targets privacy attacks in open-vocabulary, large-scale settings, where the attacker is assumed to be a deep-learning model, e.g., a multimodal foundation model or an object detection model. A privacy leakage occurs when such a model successfully detects a sensitive object in the video stream. We do not consider traditional rule-based or handcrafted attackers, as these can only recognize a fixed set of scenes.

The current probabilistic guarantee applies to the listed propositions. However, if an additional visual cue (e.g., clothing pattern) is considered privacy-sensitive, it can be explicitly added to the proposition set since our framework supports open-vocabulary detection. Regarding temporal leakage, Eq. 3 updates the probabilistic guarantee frame-by-frame but does not retroactively correct for earlier missed detections. In practice, this means that the latest probabilistic guarantee bounds cross-frame leakage or track association; i.e., the guarantee for the newest frame accounts for the potential failures of past frames.

Adversarial patches and motion blur may compromise detector confidence, leading the system to misjudge whether sensitive objects are present. While PCVS still maintains its lower-bound guarantee, the estimate

may become overly conservative under such visual corruption. This limitation can be mitigated by augmenting the calibration data with adversarial or blurred examples.

## B  Additional Experimental Settings and Results

### B.1  Benchmark Model Prompts

Recall that we use GPT-4 Vision and Video LLaVA as the benchmarks for quantitative analysis. These benchmark models require natural language input prompts to guide object detection and long-horizon object analysis.

We use the prompt below to measure how well vision-language foundation models understand the privacy specification and detect private objects or behaviors.

**Video LLaVA**  Since Video LLaVA processes a sequence of frames, we pass a video clip along with the privacy specification.

> Does this video satisfy the `<Privacy Specification>`?
> You must answer only YES or NO. For example: 'YES' or 'NO'

**GPT-4 Vision**  We pass a single frame with the prompt to the GPT-4 Vision model to identify private information specified in the privacy specification.

> Does this image satisfy the `<Privacy Specification>`?
> You must answer only YES or NO. For example: 'YES' or 'NO'

| Threshold | Iterations | Latency per frame (sec.) | PSNR |
|---|---|---|---|
| 0.50 | $31 \pm 16$ | $0.28 \pm 0.18$ | $30.50 \pm 0.02$ |
| 0.60 | $32 \pm 16$ | $0.29 \pm 0.18$ | $30.52 \pm 0.01$ |
| 0.70 | $33 \pm 16$ | $0.30 \pm 0.18$ | $30.53 \pm 0.01$ |
| 0.80 | $34 \pm 16$ | $0.31 \pm 0.18$ | $30.55 \pm 0.00$ |

Table 1: The number of iterations, latency, and PSNR remain nearly identical for different thresholds. We use an identical experimental setup as shown in Fig. 5. However, we restrict the frame length to 10.

### B.2  Additional Baseline Comparison

To represent a strong variant of traditional static redaction, we introduce the **High-Recall Static Redaction with Temporal Smoothing (HR-SR-TS)** baseline. This method operates strictly at the frame level and does not perform identity association or cross-frame reasoning. HR-SR-TS uses a high-recall detector (Bochkovskiy et al., 2020) for initial object detection, applies lightweight temporal smoothing over a short window to stabilize redaction masks, and includes selective re-detection for verification. This approach remains consistent with the classical static redaction framework since it performs no cross-frame identity association or re-identification, but improves per-frame stability through post-processing. The design is inspired by prior work on high-recall object detection and efficient online tracking (Bewley et al., 2016; Wojke et al., 2017). The pipeline therefore adheres to the classical static-redaction paradigm while improving temporal consistency through post-processing. We follow the experimental setup of Fig. 5a and evaluate on 25-frame video sequences.

| Method | Mean | Std | Min | Max | Median |
|--------|------|-----|-----|-----|--------|
| HR-SR-TS | 0.943 | 0.109 | 0.5 | 1.0 | 1.0 |
| PCVS (Ours) | **0.993** | 0.025 | 0.9 | 1.0 | 1.0 |

Table 2: Privacy preservation performance of HR-SR-TS and PCVS.

In addition to privacy metrics, we measure the visual impact of redaction using PSNR, which reflects the magnitude of non-private visual alteration. The HR-SR-TS baseline achieves a mean PSNR of 23.69 dB, while PCVS reaches 30.5 dB, demonstrating substantially improved preservation of non-private content.

**Effect under increasing concealment complexity.** We further evaluate the baselines under increased concealment complexity, corresponding to the scenario in Fig. 9, where four private objects must be concealed. As shown in Table 3, PCVS maintains significantly stronger preservation of non-private features than HR-SR-TS.

| Method | Mean | Std | Min | Max |
|--------|------|-----|-----|-----|
| PCVS (Ours) | **0.845** | 0.044 | 0.666 | 1.0 |
| HR-SR-TS | 0.686 | 0.118 | 0.000 | 0.8 |

Table 3: Performance under concealment complexity level 4.

Overall, these additional comparisons highlight the limitations of classical static redaction methods and the conceptual mismatch of DP-style systems for per-frame entity concealment. PCVS consistently provides stronger privacy protection, reduced visual distortion, and greater stability across varying concealment complexities, underscoring its suitability for real-world privacy-sensitive video applications.

### B.3 Privacy Threshold vs Visual Quality

Based on our experimental results, summarized in Table 1, the number of iterations and latency increase with an increase in the privacy threshold, which is a logical tradeoff. However, the visual quality, measured by the peak signal-to-noise ratio (PSNR) (Hore & Ziou, 2010), remains unaffected by the privacy threshold. This confirms that our concealment process is targeted and does not affect the overall quality of the image. The invariance in PSNR suggests that the concealment operates locally and selectively, avoiding unnecessary alterations to non-sensitive regions. This property is crucial for real-world applications where maintaining visual integrity is essential, such as in media publishing, video conferencing, or autonomous systems, ensuring privacy protection without compromising usability or scene comprehension. Moreover, this finding highlights the efficiency of our framework in scaling privacy protections based on user-defined thresholds while preserving the visual consistency necessary for downstream tasks.

## C  Discussion

**False Negatives**  False negatives on privacy-sensitive objects could create a misleading sense of safety. This concern directly motivates our use of a probabilistic guarantee rather than a binary privacy classification. In practice, a false negative in the detection model is a detection with very low confidence, which is then reflected in the probabilistic guarantee. Thus, the probability of missed detections is not ignored but quantitatively bounded by the same guarantee.

**Bias and Disproportionate Concealment**  Strong concealment can amplify bias when the underlying detector exhibits uneven performance across object categories or demographic groups. To address this problem, our PCVS is inherently model-agnostic and can be paired with any object-detection backbone, allowing practitioners to select or fine-tune models that meet the requirements of their deployment setting,

including fairness considerations. This flexibility enables the integration of bias-mitigated or fairness-aware detectors, thereby supporting more equitable privacy protection across diverse application domains.

