# OpenReview forum: "Real-Time Privacy Preservation for Robot Visual Perception"
_TMLR — Accepted by TMLR_

### Review · Reviewer_BGCD · 2025-08-11

**Summary Of Contributions:**

This paper introduces a method for real-time privacy preservation in robot visual perception. Their Privacy-Constrained Video Streaming (PCVS) leverages user-defined privacy specifications, object detection models, and conformal prediction to provide theoretical probabilistic guarantees that sensitive information is adequately concealed in live video streams from robots.

**Audience:**

Yes

**Audience Explanation:**

This work lies at the intersection of privacy, machine learning, and robotics. Incidents such as the cited Roomba privacy breach, along with increasing regulatory scrutiny, ensure that this topic will only become more relevant over time.

**Broader Impact Concerns:**

A broader impact statement is already included.

**Claims And Evidence:**

Yes

**Claims Explanation:**

Theoretical Claims:
- The calibration process and its necessity for bounding the confidence of object detectors is explained rigorously, with references to prior work on conformal prediction. Based on this, they claim a theoretical lower bound on privacy guarantee per frame and sequence.

Empirical Claims:
- Quantitative evaluation is conducted on three datasets (ImageNet, MS COCO, UFPR-ALPR), and across three types of robot deployments (indoor, ground robot, drone). Metrics are clearly defined (Privacy Preservation Success Ratio) and results are presented as boxplots and averages in Figures 5, 6, 8.
- CPU and GPU benchmarks are reported, showing frame rates and latencies per frame under practical conditions.
- The method’s success ratio is consistently high, and is shown to outperform GPT-4 Vision and Video-LLaVA in most scenarios .
- The authors openly report ~5% failure cases, mostly due to limitations of the object detector, and show representative examples (Figure 10b). They discuss that failures are more common on small or unusual objects (e.g., motorcycle license plates).

**Requested Changes:**

- Some evidence (especially for privacy preservation success ratios) is derived from synthetically constructed videos, not from organically captured privacy-sensitive events. This may limit generalizability and under-represent complex, real-world scenarios. Could you suggest ways in which this can be achieved or further improved?
- The benchmark needs improvement. While the detection baselines are strong, there is less comparative evidence regarding concealment quality, robustness to adversarial or challenging conditions, and direct comparison with other privacy-preserving streaming frameworks.
- In the future, it will be important to expand the benchmarks and test against models such as Gemini 2.5 Pro or GPT-4o, as these proprietary models have been found to excel at common-sense reasoning. For example, Gemini 2.5 may outperform other models in object detection due to its advanced understanding of physical plausibility. However, some recent studies on object detection using vision-language models (VLMs) suggest that their performance may not surpass that of YOLOv8 [1]. Incorporating these insights and outlining future directions would further strengthen the paper. Furthermore, although PCVS operates in real time, understanding how it compares to more accurate proprietary VLMs would help the community assess how much additional effort is needed to make the system more robust in future work.
- The prompt in Section A.1 is not comprehensive enough. Please consider using a more detailed and comprehensive prompt to evaluate whether the results can be improved. Please see Appendix B and C in [2] for examples.

[1] [Roboflow100-VL: A Multi-Domain Object Detection Benchmark for Vision-Language Models](https://arxiv.org/abs/2505.20612)
[2] [VideoGameQA-Bench: Evaluating Vision-Language Models for Video Game Quality Assurance](https://arxiv.org/pdf/2505.15952)

---

> ### Author Response · Authors · 2025-10-15
> **Response to Reviewer BGCD**
>
> Thank you for your insightful comments! Please find our responses below.
>
> *[Synthetic Data and Generalization]* We agree with the reviewer that the effect of synthetic data on the generalizability of any method must be analyzed. In the current manuscript, we include experiments with real-world, privacy-sensitive video footage from two domains: indoor robot operation and outdoor ground robot operation (see Figures 2 and 3 in our manuscript for reference). Moreover, the generalization ability of our method, PCVS, depends on the capability of the underlying vision-based object detection model to recognize objects from diverse label categories. Therefore, future work aimed at improving generalization should focus on employing detection models that can identify a broader range of object classes, for example, exploring the potential of vision-language models as the object detection model.
>
> *[Benchmarks]* We have added results for a baseline privacy-preserving method, High-Recall Static Reduction with Temporal Smoothing (HR-SR-TS), and will incorporate these results into the manuscript based on the feedback received during this discussion period. Please refer to our official comment at the top above for a comparison of our method, PCVS, with HR-SR-TS. Furthermore, **for evidence of concealment quality, please see Appendix A** of the manuscript, where we report the peak signal-to-noise ratio (PSNR), which indicates whether the concealment is targeted and quantifies its impact on the overall image quality.
>
> *[Comparison with VLMs]* We agree with the reviewer that it would indeed be interesting to compare the performance of privacy-preserving frameworks such as ours with VLMs in the future, as these models continue to evolve and improve in their reasoning and perception capabilities. However, at present, both Gemini 2.5 and GPT-4o exhibit relatively long reasoning times, making them unsuitable for real-time comparison with PCVS. Moreover, as the reviewer also notes, there is evidence that closed-set detectors may still outperform such VLMs in specific tasks. We are happy to revise our manuscript to incorporate this discussion.
>
> *[On Prompts for Baselines]* Thank you for the reference! Our rationale for the current prompts in Section A.1 is as follows: during our experiments, we observed that longer and more comprehensive prompts for the baseline VLM methods **significantly increase inference time**, causing the baselines to take substantially longer to process an entire video. Furthermore, when the prompt length is kept comparable to what we currently use in Appendix A, merely restructuring the prompt does not close the performance gap between our method and the VLM baselines (see Figure 6).

---

### Review · Reviewer_obiZ · 2025-08-23

**Summary Of Contributions:**

The paper integrates conformal prediction with live video privacy preservation, using CP to calibrate object detector outputs and produce lower bounds on detection correctness for privacy-sensitive objects. The authors introduce a video abstraction algorithm that compresses a sequence of frames into a labeled Markov chain, and, using this abstraction, provide probabilistic guarantees that a temporal logic privacy specification is satisfied across the frame sequence. They formulate a real-time privacy preservation problem, where incoming frames can be modified to ensure the abstraction satisfies the privacy rule with probabilistic guarantees.

**Strengths**
- The empirical results effectively demonstrate the utility of the proposed method. Figures 5 & 6 clearly shows superior performance in privacy preservation against baselines. Figure 9 demonstrates that non-private features are still preserved. Figure 11 highlights that using the abstraction reduces computational complexity. These empirical results support the main claims of the paper.

- The application of conformal prediction is interesting and thoughtfully executed. Propagating uncertainty from individual frames to a temporal abstraction, rather than treating each frame independently, is a clever use of domain knowledge. This approach leverages CP in a way that complements the temporal structure of the problem, rather than relying on CP alone to provide guarantees.

**Weaknesses**
- The validity of the conformal prediction guarantees relies on exchangeability of the calibration and test frames. Intuitively in a video stream, the ordering of consecutive frames is fixed, which seems to conflict with the assumption of exchangeability and may violate this assumption, affecting the validity of the theoretical lower bounds. Some discussion to clarify this key assumption would be appropriate.

- The preservation of non-private features (fig. 9) is not compared to lower bounds. This leaves open the question whether or not this methodology is simply more conservative than the baselines. I believe that this is a key piece of the empirical results that is missing.

**Additional Comments:**

I do have some background in conformal prediction, so my review focuses mostly on the CP aspects of the paper. I am not an expert in temporal logic, formal verification, or object detection, so I defer those areas to other reviewers. Where I may have misunderstood those areas, I welcome clarification from the authors.

**Audience:**

Yes

**Audience Explanation:**

This is a practically relevant topic and the authors provide a solution that spans multiple domains.

**Broader Impact Concerns:**

I do not have any broader impact concerns with this work. In fact, the methodology has the potential to positively influence broader impacts by enabling stronger privacy preservation in live video streams of robots. By reducing the risk of exposing sensitive information while maintaining utility, the approach contributes toward safer and more responsible deployment of robotic systems in real-world environments.

**Claims And Evidence:**

Yes

**Claims Explanation:**

The empirical evaluation is generally strong and supports the central claims of the paper, with the exception of the second weakness noted above. The experiments are grounded in well-chosen examples, which are presented clearly through both textual explanation and illustrative visuals. On the theoretical side, while the problem setup is rigorously defined, the justification of the exchangeability assumption could benefit from a more explicit discussion. Overall, however, the combination of empirical and theoretical contributions is convincing.

**Requested Changes:**

I believe that theses requested changes would strengthen the work.

- A more explicit discussion of the fundamental assumptions underlying the application of conformal prediction, particularly the assumption of exchangeability, would strengthen the theoretical justification of the approach.

- Figure 9 (or a related new figure/table) would benefit from comparison against a baseline. A key concern is whether stronger privacy preservation could be trivially achieved by removing all objects, which the authors briefly note on page 8 (to the left of Figure 6). However, this point warrants quantitative evidence. More generally, this reflects a fundamental type-I vs. type-II error tradeoff that should be characterized in a more formal and systematic manner.

---

> ### Author Response · Authors · 2025-10-15
> **Response to Reviewer obiZ**
>
> [*CP and exchangeability clarification*] We note that conformal prediction requires exchangeability between the calibration and test sets, meaning they must be drawn from a similar underlying distribution (i.e., i.i.d. assumption). However, CP does not require temporal independence across consecutive frames. Our objective is not to apply CP across the temporal dimension of a video. Instead, we apply CP at the **frame level** to the detection scores, and the calibration dataset is drawn from the same distribution as the test stream.
>
> Although video frames exhibit temporal correlation, this does not invalidate exchangeability, as CP guarantees remain valid under i.i.d. observations when calibration and test samples are from the same domain and share the same generative mechanism. Instead, the temporal dependencies are **explicitly modeled in the automaton abstraction layer**, where we propagate these lower bounds.
>
> [*Regarding Figure 9 and Baseline*] We agree that demonstrating preservation of non-private content is crucial. We have added quantitative results as requested by conducting a new experiment, similar to that shown in Figure 9, using the HR-SR-TS baseline with complexity level 4. The results confirm that we achieve about **16% superior performance** in preserving non-private features after concealing four private objects. More details are included in the official comment at the top: https://openreview.net/forum?id=uMf2vn8396&noteId=7kdWbIoyjD.
>  As shown in Figure 3, blurring the entire person achieves perfect privacy preservation (privacy = 1.0) but eliminates all task-relevant information, rendering the image useless for the robot. This illustrates the limitation of trivial approaches that maximize privacy without regard for utility.
>
> Moreover, Figure 9 demonstrates that our method selectively protects privacy-sensitive objects while retaining the remaining content necessary for task success. This enables a clear visualization of the type-I vs type-II privacy-utility tradeoff and shows that our method achieves strong privacy while retaining usable content, unlike trivial baselines such as HR-SR-TR. We highlight that **preserving privacy is a desirable feature of our work**. However, while doing so, we want to **ensure that it does not reduce the utility of the objects detected** in images/video streams for **downstream tasks** of the robot.

---

### Review · Reviewer_4bJk · 2025-10-05

**Summary Of Contributions:**

The paper introduces Privacy-Constrained Video Streaming (PCVS), a real-time pipeline that applies logical privacy specifications (for example, “always hide faces when a person is present”), detects objects per frame, selectively conceals a subset (blur or blackout), and maintains a running lower bound on specification satisfaction using a conformal-prediction calibration tied to a labeled Markov chain abstraction.

**Audience:**

Yes

**Audience Explanation:**

The work sits at an intersection of formal methods, robot perception, and privacy, which is of interest to readers who build real-time embodied systems and who want measurable guarantees. The incremental verification idea and the running bound are useful even beyond privacy (for example, safety rules in closed-loop control) and the study links formal specifications to modern open-vocabulary detectors.

**Broader Impact Concerns:**

The paper argues that privacy can be improved without harming robot function, which is a positive outcome. However, two risks should be discussed more directly. First, false negatives on private objects will create a false sense of safety because the running bound depends on calibration and detector coverage. Second, strong concealment may disproportionately hide certain demographics if detector bias is present.

**Claims And Evidence:**

Yes

**Claims Explanation:**

The paper reports specification satisfaction ratios across video lengths and specification complexity; shows that concealment does not break a stop-policy on the ground robot; and provides latency analysis that isolates object detection as the main cost. These results support the high-level claims that PCVS improves satisfaction rates versus baselines and is fast enough for moderate frame rates. However, the main baselines are GPT-4V and Video-LLaVA, which are not designed to deliver specification-level guarantees nor to conceal content in-loop; thus the comparison is a capability check, not an apples-to-apples privacy system baseline.

**Requested Changes:**

**Clarify calibration and composition.**
Please give a precise recipe for estimating (f_C): how the non-conformity scores are built for multi-label detections; how class-conditional versus global calibration is handled; how you prevent over-coverage when combining multiple propositions in a conjunction (product of lower bounds may be very conservative if dependencies exist).

**Threat model and “complete concealment”.**
Define the attacker and disclosure channels. Does the guarantee cover only the listed propositions, or any visual cue that could reveal the same sensitive attribute (for example, identity via clothing)? Please discuss cross-frame leakage, track association, and how missed detections in early frames affect later frames under Eq. 3. A brief discussion of adversarial patches or heavy motion blur would be useful.

**Baselines and metrics.**
Add at least one privacy-preserving video baseline that actually performs in-loop concealment with a rule checker (for example, a DP-style pipeline such as Privid or a cryptographic pipeline with proxy detection), even if it is slower, to separate “understands the rule” from “enforces the rule in real time.” Also consider a baseline with classical static redaction using a high-recall detector plus temporal smoothing, and report both satisfaction rate and redaction area (to expose over-concealment).

---

> ### Author Response · Authors · 2025-10-15
> **Response to Reviewer 4bJk**
>
> We thank the reviewer for the insightful comments, and we will try our best to address the concerns.
>
> *[Calibration and Composition]* In our framework, we construct a **global calibration set** across all object labels and build a **single non-conformity distribution** shared by all detections. This design supports **open-vocabulary privacy constraints**, where sensitive objects may not belong to a fixed label set. While class-conditional calibration can yield tighter bounds for closed vocabularies, global calibration offers better generalizability, which is a trade-off we will discuss in the revision. Regarding conjunctions, our framework intentionally provides **lower-bound guarantees**, favoring conservative estimates to ensure safety and reliability. We agree that modeling inter-object dependencies for tighter probability estimates is an important future direction and will add this discussion in our revision.
>
> *[Threat Model and Complete Concealment]* Our framework targets privacy attacks in **open-vocabulary, large-scale** settings, where the attacker is assumed to be a deep-learning model, e.g., a multimodal foundation model or an object detection model. A privacy leakage occurs when such a model successfully detects a sensitive object in the video stream. We do not consider traditional rule-based or handcrafted attackers, as these can only recognize a fixed set of scenes.
>
> The current probabilistic guarantee applies to the **listed propositions**. However, if an additional visual cue (e.g., clothing pattern) is considered privacy-sensitive, it can be explicitly added to the proposition set since our framework supports open-vocabulary detection. Regarding temporal leakage, Eq. (3) updates the probabilistic guarantee sequentially per frame but does not retroactively correct for earlier missed detections. In practice, this means that cross-frame leakage or track association is **bounded by the latest probabilistic guarantee**, i.e., the guarantee for the newest frame takes into account the potential failure of the past frames.
>
> Adversarial patches and motion blur may compromise detector confidence, leading to the system misjudging whether sensitive objects are present. While PCVS still maintains its lower-bound guarantee, the estimate may become overly conservative under such visual corruption. This limitation can be mitigated by augmenting the calibration data with adversarial or blurred examples. We will include the discussion in the revision.
>
> *[Baseline and Metrics]* We have added new video privacy-preservation baselines and demonstrated that our framework outperforms them. More details are included in the official comment at the top: https://openreview.net/forum?id=uMf2vn8396&noteId=7kdWbIoyjD.
>
> *[False Negatives]* We agree that false negatives on privacy-sensitive objects could create a misleading sense of safety. This concern directly motivates our use of a probabilistic guarantee rather than a binary privacy classification. In practice, a false negative in the detection model is a detection with very low confidence, which is then reflected in the probabilistic guarantee. Thus, the probability of missed detections is not ignored but quantitatively bounded by the same guarantee. We will clarify this mechanism in the revision and emphasize that our framework provides an uncertainty-aware estimate of compliance, i.e.,its strength lies in bounding, rather than eliminating, the residual risk.
>
> *[Bias and disproportionate concealment]* We agree that strong concealment may amplify bias if the underlying detector exhibits uneven performance. However, our framework is **model-agnostic**, i.e., it can operate with any object detection backbone, allowing practitioners to select or fine-tune the model best suited to their deployment domain and fairness requirements. In the revision, we will highlight this flexibility and include a brief discussion on choosing bias-mitigated or fairness-aware detectors to ensure equitable privacy protection across different applications.

---

### Author Response · Authors · 2025-10-15
**Baselines and Metrics for Rebuttal**

We have investigated additional baselines as per the request of multiple reviewers. **Reviewer 4bJk** suggested a few options, such as a **DP-style pipeline, classical static redaction**, and other **privacy-preserving streaming frameworks**. The most relevant DP-style video privacy-preservation work is **Privid (NSDI ‘22)**. However, there is a fundamental difference between Privid and our work. Privid does not aim to detect and blur every private object in every frame. It provides privacy guarantees over the aggregated results of video analytics queries, rather than over direct visual content. Specifically, Privid conceals entities that appear only briefly by limiting their influence on query outputs through the use of differential privacy. Hence, this baseline is not suitable for our setting.

For **classical static redaction**, we constructed a new baseline implementation termed **High-Recall Static Redaction with Temporal Smoothing (HR-SR-TS)**. This baseline follows the conventional static redaction paradigm, detecting and redacting privacy-sensitive objects independently on each frame, while introducing lightweight temporal smoothing to mitigate flicker and missed detections. Specifically, HR-SR-TS employs a high-recall detector [1] for initial frame-wise detection, applies a short-window temporal aggregation to stabilize redaction masks, and uses selective re-detection for verification. This approach remains consistent with the classical static redaction framework since it performs no cross-frame identity association or re-identification, but improves per-frame stability through post-processing. The design is inspired by prior work on high-recall object detection and efficient online tracking [1–3]. We use the same experimental setup as in Fig. 5a of the paper, but evaluate on videos with a length of 25. The full set of experiments will be included in the revised version. The probability of preserving privacy is shown in the table below.

| **Method**  | **Mean**  | **Std** | **Min** | **Max** | **Median** |
| ----------- | --------- | ------- | ------- | ------- | ---------- |
| HR-SR-TS    | 0.943     | 0.109   | 0.5     | 1.0     | 1.0        |
| PCVS (Ours) | **0.993** | 0.025   | 0.9     | 1.0     | 1.0        |

In addition, we report PSNR instead of the redaction area, since both measure the extent of visual alteration. The PSNR of our work has already been provided in the Appendix (see Table 1). The mean PSNR of the HR-SR-TS baseline is 23.69 dB, while our method achieves an average of **30.5 dB**, indicating better preservation of non-private visual content.

Lastly, following the request from **Reviewer obiZ**, we conducted the experiment shown in Figure 9 using the*HR-SR-TS baseline with complexity level 4. Our method achieves approximately **16% better performance** in preserving non-private features after concealing four private objects. Please see the results below.

| **Method**  | **Mean**  | **Std** | **Min** | **Max** |
| ----------- | --------- | ------- | ------- | ------- |
| PCVS (Ours) | **0.845** | 0.044   | 0.666   | 1.0     |
| HR-SR-TS    | 0.686     | 0.118   | 0.000   | 0.8     |

[1] [Bochkovskiy, A., Wang, C.-Y., & Liao, H.-Y. M. (2020). YOLOv4: Optimal Speed and Accuracy of Object Detection. arXiv:2004.10934](https://arxiv.org/abs/2004.10934)

[2] [Bewley, A., Ge, Z., Ott, L., Ramos, F., & Upcroft, B. (2016). Simple Online and Realtime Tracking (SORT). IEEE ICIP](https://ieeexplore.ieee.org/document/7533003)

[3] [Wojke, N., Bewley, A., & Paulus, D. (2017). Simple Online and Realtime Tracking with a Deep Association Metric (DeepSORT). IEEE ICIP](https://arxiv.org/abs/1703.07402)

---

### Decision · Action_Editor_d63X · 2025-11-11

**Recommendation:** Accept as is

**Audience:**

Yes

**Audience Explanation:**

Privacy in robotics is a relatively underexplored subject. The authors motivated their study using the Roomba privacy breach incident. As the field of robotics rapidly advances, these privacy-sensitively scenarios will become more common and addressing them will require technical solutions such as the one proposed in this paper. Thus I believe findings of this paper will likely be appreciated by the TMLR audience.

**Claims And Evidence:**

Yes

**Claims Explanation:**

The authors conduct experiments on multiple datasets, showing that their method simultaneously achieves strong utility and satisfaction of privacy specifications. Their method is also backed by conformal prediction to provide a lower bound on the detection confidence. In the rebuttal, authors also included a comparison against the HR-SR-TS baseline. Reviewers are generally convinced by the supplied evidence.

Reviewer BGCD raised weaknesses regarding evaluation on synthetic data and baseline comparison. The authors made sufficient effort to address them, and I do not believe this weakness significantly hinders the accuracy of the authors' claims.